# Proteomic Analysis of Mesenchymal Stromal Cell-Derived Extracellular Vesicles and Reconstructed Membrane Particles

**DOI:** 10.3390/ijms222312935

**Published:** 2021-11-29

**Authors:** Hector Tejeda-Mora, Leticia G. Leon, Jeroen Demmers, Carla C. Baan, Marlies E. J. Reinders, Bertram Bleck, Eleuterio Lombardo, Ana Merino, Martin J. Hoogduijn

**Affiliations:** 1Erasmus MC Transplant Institute, Department of Internal Medicine, Erasmus University Medical Center, 3015 GD Rotterdam, The Netherlands; h.tejedamora@erasmusmc.nl (H.T.-M.); c.c.baan@erasmusmc.nl (C.C.B.); m.e.j.reinders@erasmusmc.nl (M.E.J.R.); merino.a76@gmail.com (A.M.); 2The Netherlands Cancer Institute, Department of Pathology, Plesmanlaan 121, 1066 CX Amsterdam, The Netherlands; l.gleon@nki.nl; 3Proteomics Center, Erasmus MC, Erasmus University Medical Center, 3015 GD Rotterdam, The Netherlands; j.demmers@erasmusmc.nl; 4Takeda, Gastrointestinal Drug Discovery Unit, Cambridge, MA 02139, USA; bertram.bleck@takeda.com; 5Takeda Madrid, Cell Therapy Technology Center-Cell Therapies, 28046 Madrid, Spain; Eleuterio.Lombardo@takeda.com

**Keywords:** membrane particles, mesenchymal stromal cell, extracellular vesicle, proteomics

## Abstract

Extracellular vesicles (EV) derived from mesenchymal stromal cells (MSC) are a potential therapy for immunological and degenerative diseases. However, large-scale production of EV free from contamination by soluble proteins is a major challenge. The generation of particles from isolated membranes of MSC, membrane particles (MP), may be an alternative to EV. In the present study we generated MP from the membranes of lysed MSC after removal of the nuclei. The yield of MP per MSC was 1 × 10^5^ times higher than EV derived from the same number of MSC. To compare the proteome of MP and EV, proteomic analysis of MP and EV was performed. MP contained over 20 times more proteins than EV. The proteins present in MP evidenced a multi-organelle origin of MP. The projected function of the proteins in EV and MP was very different. Whilst proteins in EV mainly play a role in extracellular matrix organization, proteins in MP were interconnected in diverse molecular pathways, including protein synthesis and degradation pathways and demonstrated enzymatic activity. Treatment of MSC with IFNγ led to a profound effect on the protein make up of EV and MP, demonstrating the possibility to modify the phenotype of EV and MP through modification of parent MSC. These results demonstrate that MP are an attractive alternative to EV for the development of potential therapies. Functional studies will have to demonstrate therapeutic efficacy of MP in preclinical disease models.

## 1. Introduction

Mesenchymal stromal cells (MSC) play an important role in immunomodulatory and regenerative processes by interacting with a variety of immune and progenitor cell types. A major part of these interactions are mediated via the MSC secretome, which includes a range of soluble immune and trophic mediators [1] and extracellular vesicles (EV). When they were first identified, EV were shown to play a role in controlled shedding of factors to the extracellular space [2], but it has now become clear that EV also contain functional proteins, microRNAs and even depolarized mitochondria, and carry signals to target cells [3,4,5]. MSC actively secrete EV that target, amongst others, immune and progenitor cells [6,7]. MSC-derived EV have for instance been shown to polarize inflammatory macrophages to regulatory macrophages [8], induce apoptosis of subsets of T lymphocytes [9] and stimulate neuron branching and outgrowth [10]. In addition to their biological effects, MSC-derived EV are reported to modulate disease onset and progression in, amongst others, models of acute kidney injury [11,12], myocardial ischemia [13] and lung injury [14] by MSC-derived EV.

Notwithstanding the promising and potentially versatile therapeutic effects of EV, the development of EV as a medical product faces a number of major hurdles, not in the first place due to the difficulty of generating large amounts of EV free from contamination with cell-derived soluble factors [15,16] and components from the culture medium, such as serum-derived RNAs [17]. Furthermore, EV, even derived from one cell type, are heterogeneous in protein content [18] and therefore it is difficult to produce EV with a consistent protein make-up. EV are produced and secreted via poorly understood mechanisms, and therefore currently it is unknown how to modify the content and membrane protein composition of EV. Vesicle-like structures that can be produced at a large scale in a controlled manner, isolated free from contamination and that allow modification of their molecular setup would offer major benefits for the development of vesicle-based therapies.

We recently developed methodology to isolate and reconstruct the membranes of adipose tissue derived MSC [19]. These membrane particles (MP) are generated from MSC after washing the cell culture supernatant, subsequent lysis of the cells, removal of the nuclei and extrusion of the collected membranes through a 200 nm pore size filter. The MP generation procedure avoids co-precipitation of soluble factors and allows the generation of around 1 × 10^5^ MP per MSC. We demonstrated earlier that MP are potent modulators of monocyte function [19] and in this sense MP mimic the function of MSC [20]. Multiple studies have demonstrated that stimulation of MSC with pro-inflammatory cytokines such as IFNγ and TNFα affects the phenotype and function of MSC [21,22,23]. The proteomic changes that MSC undergo in response to these stimulations are well described and it is expected that these changes directly affect the protein make-up of MP. The proteome of MP is however unknown and it is unclear how challenging MSC with pro-inflammatory cytokines modifies the MP proteome. Furthermore, it is unknown how the proteome and biological function of MP compares to EV in a direct comparison.

In the present study we generated EV and MP from the same MSC cultures and compared their protein make-up by proteomic analysis. In addition, the protein content of EV and MP generated from MSC treated with IFNγ was examined and variation in EV and MP proteome between different donors determined. Finally, we determined the enzyme activities of MP and EV.

## 2. Results

### 2.1. Generation of Extracellular Vesicles and Membrane Particles from MSC

To generate EV and MP, human adipose tissue-derived MSC were expanded in culture in the presence or absence of IFNγ. The cells showed a typical spindle-shaped morphology and the majority expressed the markers CD13, CD73, CD90, HLA class I, and were negative for CD31, CD45, HLA class II and PD-L1 (data not shown). IFNγ-stimulated MSC showed increased expression levels of HLA class I and II, and PD-L1. EV were obtained from 24 h MSC conditioned medium, and MP were generated from trypsinised MSC (Figure 1). While it is known that EV are circular and exhibit a lipid double membrane, we performed electron microscopy to determine MP morphology. MP demonstrated a circular morphology with a double lipid membrane (Figure 2), similar to EV. The concentration and size of EV and MP were analyzed by Nanoparticle tracking analysis. EV showed an average size of 168 ± 19 nm and MP of 123 ± 13 nm. To compare production efficiency, the number of EV and MP generated per cell was calculated. On average 3.7 EV per MSC were collected. The number of MP generated per MSC was 4–5 magnitudes higher with 1 × 10^5^ MP per MSC (Table 1).

### 2.2. Membranes Particles Show More Diverse Proteome Than Extracellular Vesicles

Mass spectrometry was performed to compare the proteome of EV and MP. Only proteins that were detected in EV or MP samples from at least two out of three donors were taken into consideration. EV expressed 78 different proteins and treatment with IFNγ introduced 24 new proteins in EVγ, whereas 15 proteins disappeared (Figure 3A). A total of 23 proteins increased expression in EV upon IFNγ treatment of MSC up to a maximum of 4.7-fold increase. Decreased expression of 42 proteins was observed, with a maximum fold decrease of 1.9. MP expressed 1637 proteins, 21 times more than EV, of which 42 proteins overlapped with EV. Treatment with IFNγ introduced 147 new proteins in MP and caused a loss of 149 proteins. There was an increase in expression of 662 proteins in MP upon IFNγ treatment of the MSC, whereas 661 proteins decreased expression. A list of all proteins represented in the Venn diagrams is shown in Appendix A.

### 2.3. Intracellular Origin of Proteins in Extracellular Vesicles and Membrane Particles

Proteins present in EV were for a large extent functionally associated with the extracellular matrix (51 out of 78 proteins) (Figure 3B). Furthermore, 35 proteins were associated with the plasma membrane and 31 proteins with the endoplasmic reticulum, whereby it has to be taken into consideration that proteins can be associated with multiple cellular compartments. The proteins in MP originated from all cellular organelles, including ribosomes (87 proteins), the plasma membrane (436 proteins), mitochondria (244 proteins), the Golgi apparatus (178 proteins), the endoplasmic reticulum (260 proteins), the cytoskeleton (316 proteins), and cytoplasmic vesicles (402 proteins). A large number of proteins in MP were associated with the nucleus and with the nucleoplasm (787 and 413 proteins respectively). As nuclei are removed before generation of MP, these proteins are likely present in the endoplasmic reticulum, the Golgi apparatus or associated with microtubule transport networks before they are transported to the nucleus. When comparing the origin of proteins after normalizing for the different numbers of proteins in MP and EV, we can observe that EV are in particular enriched for extracellular matrix proteins, endoplasmic reticulum and cytoplasmic vesicle proteins compared to MP, whereas MP are enriched for ribosomal proteins, nucleus, nucleoplasm and mitochondrial proteins.

### 2.4. Contamination of Extracellular Vesicles and Membrane Particles with Soluble Proteins

There was no evidence for the presence of soluble factors that are abundantly expressed by MSC such as vascular endothelial growth factor (VEGF), IL6, IL8, tissue inhibitor of metalloproteinases 1 and 2 (TIMP1 and TIMP2), and low abundance of secreted protein acidic and rich in cysteine (SPARC) in the MP samples, suggesting MP are relatively free of contamination from soluble factors and mainly contain membrane or cytoskeleton associated proteins. EV were also free from VEGF, IL6 and IL8, but contained TIMP1, TIMP2 and SPARC, confirming reports that ultracentrifugation cannot clear EV of all soluble factors [15,16].

### 2.5. IFNγ Treatment Affects MP Proteome

To examine the variation in protein composition in EV and MP from different MSC donors and the effect of IFNγ treatment, principal component (PC) analysis was performed and coefficient of variability (CV) values for inter and intra sample variation for all proteins calculated. EV from different donors strongly clustered together with 57% intra sample CV, indicating that variation between EV from different donors is relatively small (Figure 4A). MP showed larger donor variation (68% CV), but clustered together separately from EV (59% inter sample CV for all proteins). On this global scale, IFNγ treatment appeared to have little effect on EV clustering, while MPγ clearly clustered separately from MP. Hierarchical clustering confirmed the difference between EV and MP, but also indicated an effect of IFNγ in both groups (Appendix A). Zooming in on the effects of IFNγ on EV and EVγ samples demonstrated that 5 proteins were significantly downregulated and 14 proteins that were upregulated in EVγ vs EV (Figure 4B, and Appendix A). These included in particular proteins with a function in the complement system and proteins that play a role in extracellular matrix remodeling. Hierarchical clustering separated MP from MPγ (Appendix A) and zooming in on the differentially expressed proteins revealed 11 proteins that were significantly downregulated in MPγ compared to MP, and 41 proteins that were upregulated (Figure 4C, Appendix A). The downregulated proteins were diverse in intracellular origin and function and included five proteins of mitochondrial origin and four involved in extracellular matrix formation. Among the upregulated proteins were HLA class I and II molecules, intercellular adhesion molecule 1 (ICAM1), indoleamine 2,3-dioxygenase 1 (IDO1) and guanylate binding proteins, which are known to be upregulated by MSC upon immune activation. Furthermore, IFN-responsive proteins were upregulated in MP derived from MSC that were treated with IFNγ, indicating that the IFNγ-induced phenotype of MSC is translated in MP.

### 2.6. IFNγ Treatment of MSC Affects Molecular Pathways in Extracellular Vesicles and Membrane Particles

Several of the proteins present in EV and MP are functionally connected in molecular pathways. To analyse whether IFNγ-treatment affected the prevalence of pathways in EV and MP, we determined the enrichment of proteins within these pathways. In EVγ, in particular protein in the pathway mediating extracellular matrix organization were more abundantly present (Figure 5A, and Appendix A. The pathways most significantly enriched in MP upon IFNγ treatment of MSC were protein degradation and ribosomal pathways. As the effect of IFNγ on proteasome and ribosomal proteins was very significant, other pathways affected by IFNγ were obscured and therefore we removed proteasome and ribosomal proteins from the analyses. This revealed that tRNA aminoacylation and IFNγ-stimulated anti-viral pathways were significantly enhanced in MPγ compared to MP (Figure 5B and Appendix A).

### 2.7. Membrane Particles Possess Enzyme Activity

In addition to the effects of EV and MP through uptake by target cells, they may also mediate biological effects independent of a target cell via the enzymatic activity of proteins in their membranes. We found the presence of 8 ATPases on MP but not on EV and detected the presence of ecto-5′-nucleotidase (CD73) on MP (Figure 6A). To analyze whether these proteins were functionally active, we measured the ability of MP, MPγ, EV and EVγ to convert ATP to ADP through ATPase activity. We found robust ATPase activity in MP, but not in EV (Figure 6B). In addition, we detected ecto-5′-nucleotidase (CD73) activity through the conversion of AMP to adenosine by MP, but not EV (Figure 6C). IFNγ-treatment did not affect the ATPase and ecto-5′-nucleotidase activity of MP. These results demonstrate that proteins found on MP are functionally active.

## 3. Discussion

EV are recognized as a promising, but poorly understood, potential therapeutic product for treatment of inflammatory and degenerative disease [24,25]. MSC constitutively secrete EV and are relatively easy to isolate and expand to large numbers in culture and may therefore be applied as EV factories [26]. However, the collection of therapeutic amounts of MSC derived EV, free from contamination by soluble factors, is a big challenge. Furthermore, we have currently little knowledge on how to modulate the phenotype and function of MSC derived EV. We therefore generated particles from the membranes of human adipose tissue derived MSC by lysing MSC, removal of the nuclei, and subsequent rearranging of the collected membranes through extrusion. The yield of these reconstructed membrane particles (MP) per MSC was 25,000 times higher than of EV, which in terms of quantity, positions MP as a more feasible product for therapy development than EV.

To characterize MP and aid their investigation for therapeutic use, we examined the protein make up of MP and made a comparison with MSC-derived EV. There was some degree of overlap in the protein make up of MP and EV, but both MP and EV also contained several unique proteins. An important observation was that compared to EV, there was little contamination of MP by soluble proteins. This suggests that the MP generation procedure is efficient in washing away soluble proteins, whereas the EV isolation method does not efficiently eliminate soluble proteins. Even though the same amount of peptide was loaded on the mass spectrometer, the number of different proteins that were detectable in MP was more than 20 times higher than in EV, indicating that MP have a much more diverse proteome than EV. The reason for the difference in the number of different types of protein between EV and MP is that MSC-derived EV are relatively homogeneous, whereas MP originate from the plasma membrane and from membrane containing organelles and contain much of the cellular protein machinery that is embedded in the various membranes. Whether MP are a homogenous population containing mixed membranes derived from diverse organelles, or whether there are subsets of MP that are entirely composed of membranes of a particular organelle, we cannot extract from the data of the present study.

Several of the proteins in MP and EV were connected through putative molecular signaling pathways. The molecular pathways in EV were mainly involved in extracellular matrix organization. This observation fits recent insights that EV are structural and functional components of the extracellular matrix [27,28,29]. The most significant pathways in MP were involved in protein synthesis, including initiation of translation and translation elongation. Preliminary data indicates that MP contain RNA (data not shown) and, speculatively, the presence of the protein synthesis apparatus in MP may allow protein synthesis of the present mRNA molecules by MP. This concept requires further investigation in follow up studies.

IFNγ stimulation of MSC had an impact on the proteome of EV and MP. In EV derived from IFNγ-stimulated MSC further enhancement of pathways involved in extracellular matrix organization was observed. In MP there was a broad impact of IFNγ. It is known that IFNγ has a major impact on several molecular pathways and affects different functional properties of MSC, and as MP are composed of membranes of diverse origins it is expected that IFNγ treatment of MSC leaves a multifaceted footprint on MP. IFNγ upregulates a range of anti-inflammatory molecules in MSC, elevates HLA class I expression and induces HLA class II expression [22,30,31]. We found in the first place an up regulation of proteasome pathways in MP upon IFNγ treatment of MSC. It has been known for a long time that IFNγ induces proteasome activity in immune cells and thereby stimulates their antigen presentation activity [32,33]. MSC have been shown to possess antigen-presenting capacity, which is increased upon treatment of MSC with IFNγ [34]. This finding is supported by the increase in proteasome pathways in MP derived from IFNγ-treated MSC. Upon uptake of MP by immune cells the proteasome activity of MP may affect antigen presentation in these cells and thereby modulate immune responses.

Underlying the increase in proteasome pathways in MP upon IFNγ treatment of MSC, we observed an increase in the already abundantly present pathways involved in protein synthesis and there was an increase in IFNγ-induced antiviral pathways and endocytosis pathways. These results demonstrate that selective treatment of MSC can modulate the proteome make up of MP. This offers the possibility to treat MSC with factors that induce specific properties in MP. In this way tailored MP can be generated for treatment of specific disorders.

Finally, to demonstrate that the proteins in MP were functionally intact, we measured enzymatic activity of MP. MP contained eight types of ATPases, which were of plasma membrane, endoplasmic reticulum and lysosomal origin. Furthermore, MP contained ecto-5′-nucleotidase (CD73). These proteins were not detectable in EV. MP were able to convert ATP to ADP and AMP to adenosine via these enzymes. These results demonstrate that the MP generation procedure leaves proteins intact. Furthermore, ATP depletion and adenosine production are implicated in immunomodulation and MP may therefore be able to locally modulate the microenvironment into an immunosuppressive state.

The results of the present study demonstrate that MSC-derived MP are an alternative to MSC-derived EV as a potential therapeutic agent. MP can be produced in far higher quantity free from contamination, they possess a much broader proteome, and exhibit enzymatic activity. Modification of MSC protein expression leaves an imprint on the proteome of MP, and eventually this may offer the possibility to generate modified MP for specific disorders. Ongoing research will further explore the functionality of MP and help development of MP therapy.

## 4. Materials and Methods

### 4.1. Mesenchymal Stromal Cell Isolation and Expansion

MSC were isolated from abdominal subcutaneous adipose tissue of healthy volunteers during a kidney donation procedure after obtaining written informed consent, as approved by the Medical Ethics Committee of the Erasmus University Medical Centre Rotterdam (protocol No. MEC-2006-190). Within hours after collection, adipose tissue was mechanically disrupted and dissociated by treatment with 0.5 mg/mL collagenase type IV (Life Technologies, Paisley, UK) in RPMI 1640 (Life Technologies) for 30 min at 37 °C under continuous shaking. After washing, the stromal vascular cell fraction was cultured in minimum essential medium Eagle alpha (MEM-α; Sigma Aldrich, St. Louis, MO, USA) with 2 mM L-glutamine (Lonza, Verviers, Belgium) and 1% penicillin/streptomycin solution (P/S; 100 IU/mL penicillin, 100 IU/mL streptomycin; Lonza, Verviers, Belgium) at 37 °C, 5% CO_2_, and 20% O_2_. MSC appeared as plastic adherent, spindle shaped cells and were passaged when reaching 90% confluency using 0.05% trypsin-EDTA (Life Technologies, Bleiswijk, The Netherlands). MSC were cultured until passage 5–6.

### 4.2. Treatment of Mesenchymal Stromal Cells with IFNγ

To obtain IFNγ-challenged MSC, the cells were cultured with 50 ng/mL IFNγ (Sigma-Aldrich, St. Louis, MO, USA) for 72 h. Medium was then removed, the cells washed with PBS, and EV isolated and MP generated as described below and as shown in Appendix A.

### 4.3. Immunophenotypic Characterization of Mesenchymal Stromal Cells

Unstimulated and IFNγ-stimulated MSC were incubated with mouse-anti-human monoclonal antibodies against CD13-PE-Cy7; CD45-APC; HLA-DR-PERCP; HLA-ABC-APC; CD31-FITC; CD73-PE; PD-L1-PE (all BD Biosciences, San Jose, CA, USA); and CD90-APC (R&D Systems, Abingdon, UK) at room temperature in the absence of light for 30 min. After two washes with FACS Flow, flow cytometric analysis was performed using FACSCANTO-II with KALUZA Software (BD, San Jose, CA, USA).

### 4.4. Isolation of Extracellular Vesicles

MSC of three donors were cultured in three T175 cm^2^ flasks until 90% confluency with or without IFNγ. They were then washed with PBS and cultured for 24 h in 15 mL serum-free medium (MEM-α). Conditioned medium (45 mL) was then collected and floating cells and cellular debris removed by centrifugation at 300× *g* for 30 min. Subsequently, the supernatant was ultracentrifuged in polyallomer centrifuge tubes (Beckman Coulter) at 100,000× *g* for 2 h using a Beckman Coulter ultracentrifuge (Beckman Coulter Optima L-90K ultracentrifuge; Beckman Coulter, Fullerton, CA, USA) with a swing angle rotor type SW40Ti, similar as described before [35]. EV were collected in 200 μL of filtered PBS and stored at −80 °C until use.

### 4.5. Generation of Membrane Particles

MSC of three donors were cultured in T175 cm^2^ flasks until 90% confluency and removed from the culture flasks by trypsinisation. The MSC were then incubated in milliQ water at 4 °C to induce osmotic lysis and after about 20 min liberation of the cell nuclei was observed under the microscope. Cell extracts were cleared of unbroken cells and nuclei by centrifugation at 2000× *g* for 20 min. The supernatant was transferred to Amicon Ultra-15 filter tubes of 100 kDa pore size and centrifugated at 4000× *g* at 4 °C. The obtained pellet consisting of crude membranes and organelles was dissolved in 0.2 µm-filtered PBS and extruded through polycarbonate membrane filters (Merck KGaA, Darmstadt, Germany) with a pore diameter of 800 nm, 400 nm and finally 200 nm using a LiposoFast LF-50 extruder (AVESTIN Europe, Mannheim, Germany) at 20 psi. All procedures were performed on ice. The obtained MP were stored at −80 °C.

### 4.6. Nanoparticle Tracking Analysis (NTA)

Analysis of concentration and size distribution of EV and MP was performed by NanoSight NS300 (Malvern Instruments, Malvern, UK) using the following settings: detection threshold 3, three measurements per sample (30 s/measurement), temperature 23.61 ± 0.8 °C; viscosity 0.92 ± 0.02 cP, 25 frames per second. Samples were diluted to a measurable concentration of particles (1 × 10^8^ particles/mL) in accordance with the manufacturer’s recommendations.

### 4.7. Proteomic Analysis (Mass Spectrometry)

EV were isolated from the conditioned medium of MSC (24 h, serum-free conditions), and MP were subsequently generated from the same MSC, as described above. EV and MP were lysed in an ice-cold buffer containing 100 mM Tris-HCl (pH 8.5), 12 mM sodium DOC and 12 mM sodium N-lauroylsarcosinate. The lysate was sonicated for 10 min in a Diagenode Bioruptor at 4 °C and then heated for 5 min to 95 °C. Proteins were subjected to reduction with dithiothreitol, alkylation with iodoacetamide and then in-solution digested with trypsin (sequencing grade; Promega, Madison, WI, USA). Proteolytic peptides were collected, washed and equal amounts of peptide were loaded and analyzed by liquid chromatography tandem mass spectrometry (nLC-MS/MS) performed on an EASY-nLC coupled to an Orbitrap Fusion Lumos Tribid mass spectrometer (Thermo Fisher Scientific, Rockford, IL, USA) operating in positive mode. Peptides were separated on a ReproSil-C18 reversed-phase column (Dr Maisch, Ammerbuch, Germany; 15 cm × 50 μm) using a linear gradient of 0–80% acetonitrile (in 0.1% formic acid) during 90 min at a rate of 200 nL/min. The elution was directly sprayed into the electrospray ionization (ESI) source of the mass spectrometer. Spectra were acquired in continuum mode; fragmentation of the peptides was performed in data-dependent mode by HCD.

Raw mass spectrometry data were analyzed with the MaxQuant software suite [36] version 1.6.7.0 with the additional options ‘LFQ’ and ‘iBAQ’ selected. The Andromeda search engine was used to search the MS/MS spectra against the Uniprot database (taxonomy: *Homo sapiens*, release: June 2019) concatenated with the reversed versions of all sequences. A maximum of two missed cleavages was allowed. The peptide tolerance was set to 10 ppm and the fragment ion tolerance was set to 0.6 Da for HCD spectra. The enzyme specificity was set to trypsin and cysteine carbamidomethylation was set as a fixed modification. Both the PSM and protein FDR were set to 0.01. In case the identified peptides of two proteins were the same or the identified peptides of one protein included all peptides of another protein, these proteins were combined by MaxQuant and reported as one protein group. Before further statistical analysis, known contaminants and reverse hits were removed. All downstream analyses such as statistical t testing, etc. were performed with the Perseus module of the MaxQuant software suite.

### 4.8. Bioinformatics Analysis

Bioinformatics analyses were performed in R (version 4.0.4, R Core Team 2020, Vienna, Austria). Protein classification was done by mapping gene IDs to the human terms using the webtool Gene Ontology. Unannotated genes and genes with non-root annotations were left out for classification. The dataset contained proteins which were not quantified in all samples. Missing values were treated as data missing not at random (MNAR) due to experimental conditions, and these were imputed by random draws from a left-shifted Gaussian distribution. Differential enrichment analysis was performed with DEP (version 1.15.0) [37] based on protein-wise linear models and empirical Bayes statistics.

Pathway identification and pathway enrichment analysis were performed with pathfindR (version 1.6.1) [38] using the Reactome gene set (version 70) and the Biogrid protein-protein interaction database (version 1.1.1). Briefly, proteins (gene IDs) with their associated *p*-values and log-fold-change values were mapped onto the Reactome gene set. Search of active pathways was performed, and the resulting active pathways were sorted based on their pathway score (quartile threshold = 0.80) and the number of genes they contained (threshold = 2). Enrichment analyses were then performed using the gene IDs in each of the active pathways. To determine whether certain genes are enriched, the hypergeometric distribution was assumed and *p*-values are calculated. Adjustment of *p*-values was done using the Bonferroni method. Enrichment results were then filtered by an adjusted *p*-value threshold of 0.05. Variation among MP and EV samples was assessed using principal component analysis (PCA) and hierarchical clustering using average linkage; *p*-values were computed for each of the clusters via multiscale bootstrap resampling (*n* = 10,000). Furthermore, %CV values between identified proteins among samples were calculated. Proteins that were present in at least two out of three samples were taken into consideration for ontology, enrichment and pathway analysis. For donor variation analysis all proteins were taken into account.

### 4.9. ATPase Assay

ATPase activity from MP, MPγ, EV and EVγ was measured using an ATPase assay kit according to the manufacturer’s instructions (Sigma-Aldrich). A phosphate standard was used for creating a standard curve. MP, MPγ, EV and EVγ (1 × 10^11^ particles/mL) were incubated with 4 mM ATP for 60 min at room temperature in assay buffer with malachite green reagent. The formation of the colorimetric product that forms in the presence of free phosphates was measured with a spectrophotometer at 620 nm. As a control for possible phosphate contamination, samples were incubated in assay buffer without ATP. The signal from these samples was subtracted from the measurements.

### 4.10. Ecto-5′-Nucleotidase Activity Assay

Ecto-5′-nucleotidase activity of MP, MPγ, EV and EVγ (1 × 10^11^ particles/mL) was analyzed by colorimetric ecto-5′-nucleotidase inhibitor screening assay, whereby ecto-5′-nucleotidase from the assay kit was replaced by MP or EV sample, according to the manufacturer’s instructions (BPS Bioscience, San Diego, CA, USA).

## Figures and Tables

**Figure 1 ijms-22-12935-f001:**
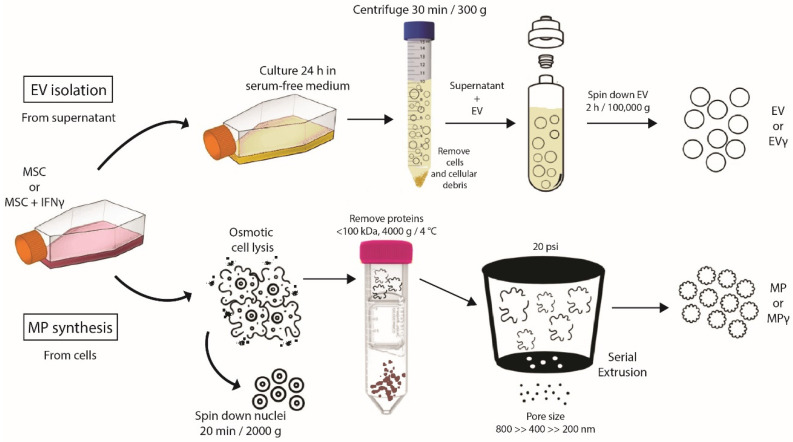
Schematic overview of EV, EVγ, MP and MPγ generation.

**Figure 2 ijms-22-12935-f002:**
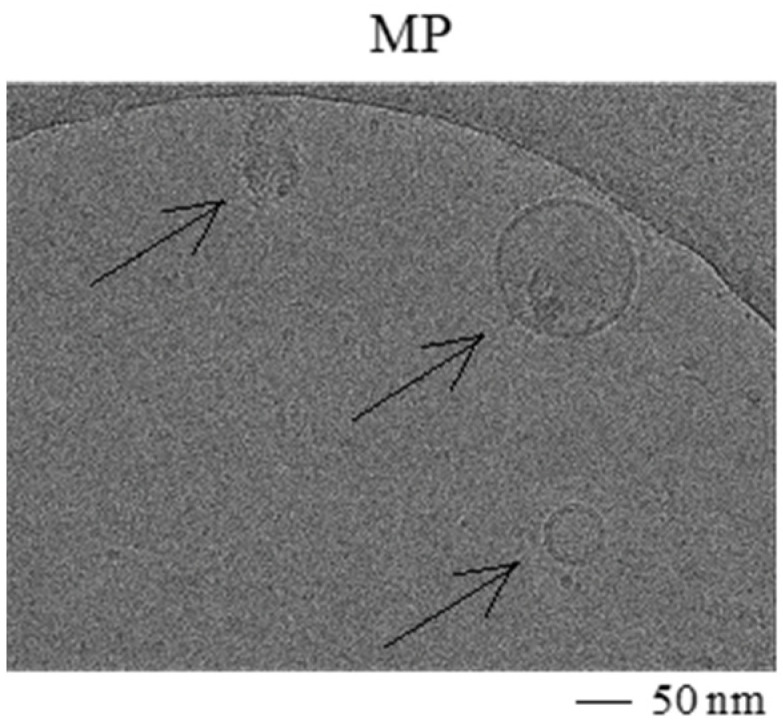
Electron microscopy image of MP. Images demonstrate a lipid bilayer in MP.

**Figure 3 ijms-22-12935-f003:**
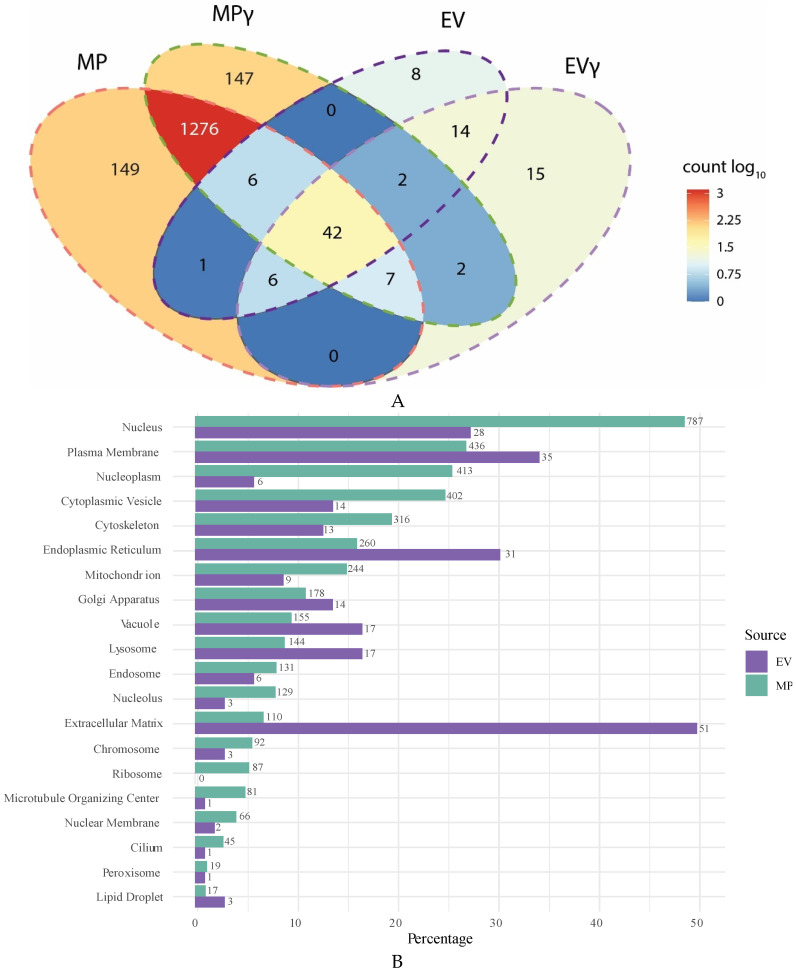
Proteomic comparison of EV, EVγ, MP and MPγ. (**A**): Venn chart showing the number of proteins expressed in EV, EVγ, MP and MPγ and overlap in expression between the different vesicle types. (**B**): Normalized comparison of the origin of EV and MP proteins. Bars depict the percentage of proteins associated with the indicated cellular compartment, numbers indicate the number of proteins that can be associated with the diverse cellular compartments.

**Figure 4 ijms-22-12935-f004:**
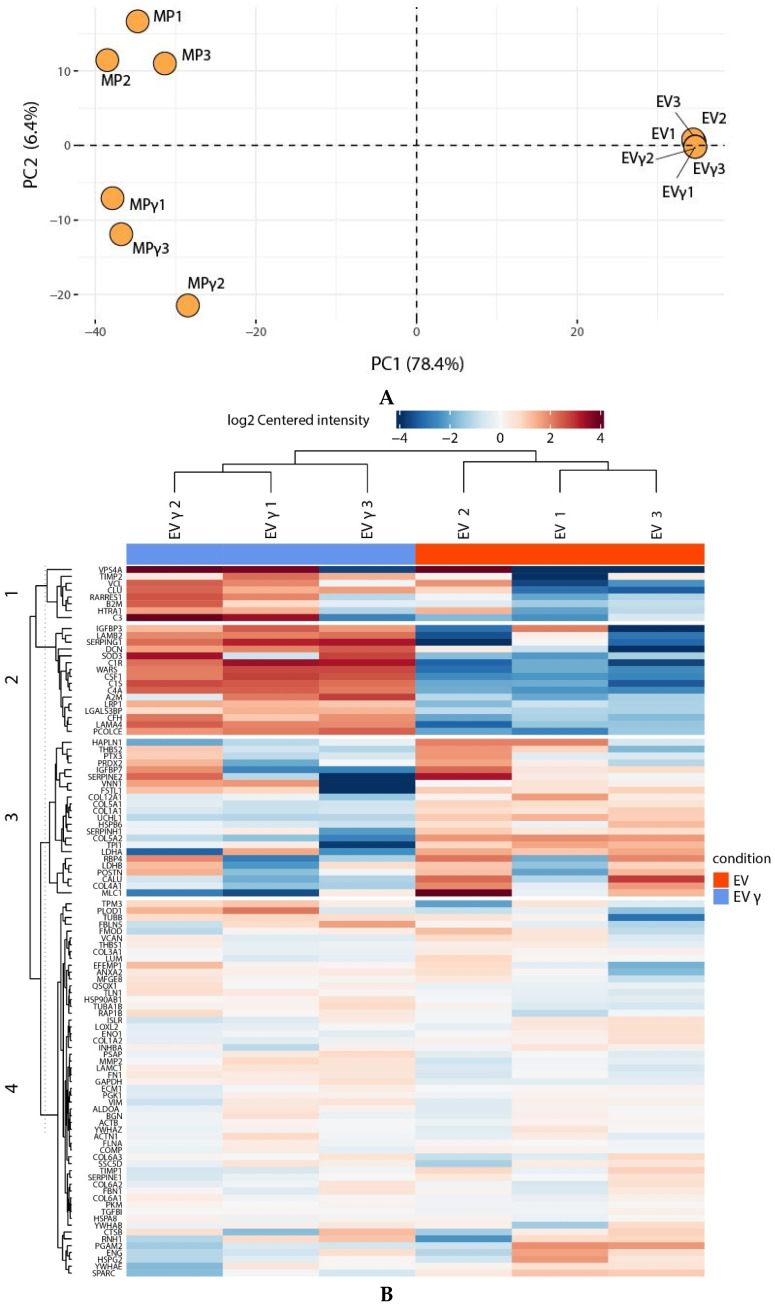
Effect of IFNγ treatment of MSC on EV and MP proteome. (**A**): Principle component analysis of EV, EVγ, MP and MPγ from three MSC donors on normalized abundance. The plot illustrates discrete categorization of biological replicates with relatively little donor variation in EV and larger donor variation in MP in PC1. (**B**): Heatmap with all significant enriched proteins in all EV samples (*p*-value ≤ 0.05). Hierarchical clustering using average linkage shows that IFNγ has a strong effect on the protein composition of EV. (**C**): Heatmap with all significant enriched proteins in all MP samples (*p*-value ≤ 0.05, log2 fold-change > 1.5). Hierarchical clustering using average linkage shows that IFNγ has a strong effect on the protein composition of MP.

**Figure 5 ijms-22-12935-f005:**
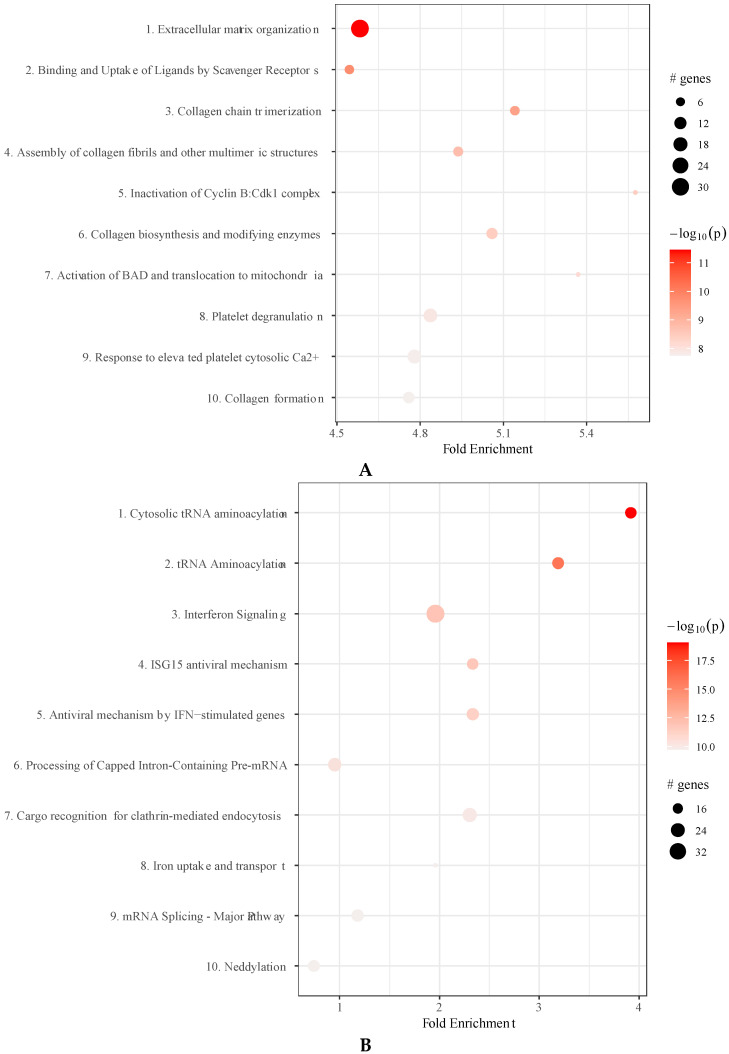
Pathway analysis. (**A**): Bullet chart displaying enriched pathways in EVγ compared to EV. Size of the bullets indicates the number of proteins in the given enriched pathway. Color indicates the −log10(lowest *p*-value). (**B**): Bullet chart displaying enriched pathways in MPγ compared to MP after removal of proteasome and ribosome related pathways. Size of the bubble indicates the number of proteins in the given enriched pathway. Color indicates the −log10(lowest *p*-value).

**Figure 6 ijms-22-12935-f006:**
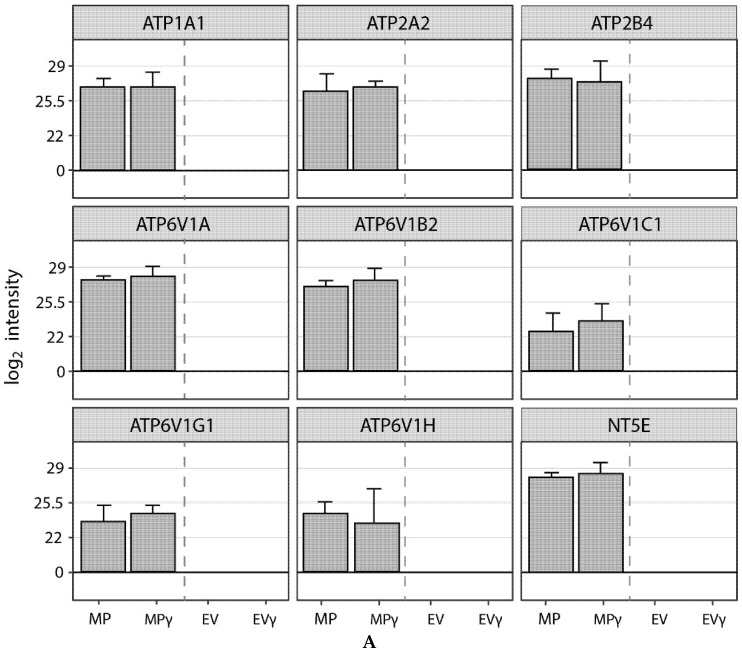
Enzymatic activity of EV and MP. (**A**): Relative abundance of 8 types of ATPases and ecto-5′nucleotidase in EV, EVγ, MP and MP. (**B**): ATPase activity was measured at a concentration of MP and EV of 1 × 10^11^ particles/mL. MP and MPγ catalyzed the breakdown of ATP at similar efficiency, whereas no activity was not observed for EV. (**C**): Ecto-5′-nucleotidase activity was measured at a concentration of MP and EV of 1 × 10^11^ particles/mL. MP and MPγ catalyzed the breakdown of AMP at similar efficiency, whereas no activity not observed for EV.

**Table 1 ijms-22-12935-t001:** MSC donor characteristics. Characteristics of MSC donor age, gender, MSC passage number and number of EV and MP isolated per MSC.

Donor No.	Gender	Age	MSC Passage	EV/MSC	MP/MSC
1	Female	58	6	5.0	11 × 10^4^
2	Male	34	6	4.8	9.6 × 10^4^
3	Male	69	5	1.3	8.8 × 10^4^

## Data Availability

The data presented in this study are available on request from the corresponding author.

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
