# Peer review of "Proteomic Analysis of Mesenchymal Stromal Cell-Derived Extracellular Vesicles and Reconstructed Membrane Particles"

_ijms, 2021, doi:10.3390/ijms222312935_

Round 1
Reviewer 1 Report
The manuscript of Tejeda and co-workers reports a proteomic analysis due in MPs and EVs derived from MSCs and in these samples after INFγ treatment. The topic of the article is very interesting, but some changes will be needed for publication:
- The authors need to include as a supplementary material the list of all the proteins detected in all the conditions tested.
- Authors mentioned in the introduction that there bibliography describing proteome of MSC but especially in the case of MPs that are generated from MSCs It will be very valuable a comparative study between the whole cell and the MPs and after including EVs proteome study, therefore, they are vesicles naturally formed from MSCs
- It will be valuable to include the full list of the 42 common proteins between MPs and EVs
- In figure 4 at size 100% is impossible to read the code of the proteins
- Authors claim that they demonstrate that the proteins in MP were functionally intact but they measured enzymatic activity of MP. They only measured ATPase activity and It will be valuable to include other activities related to other proteases or even the inhibitory activity of the MPs that it seems that authors detect some protease inhibitors in these vesicles.
- Authors talk about a more diverse proteome in MPs but they need to clarify that there are a lot of MPs, and that they contain a higher quantity of proteins than EVs.
Author Response
Dear Dr. Albert Li,
We thank you for considering our manuscript for publication in the International Journal of Molecular Sciences and many thanks also to the reviewers who took time to read our work. The requested revisions are highlighted in the manuscript and we believe that these have strengthened the manuscript. Please find below a point-by-point description of the changes we made in the document and the answers to the questions raised by the reviewers.
We hope the revised manuscript is acceptable for publication in the International Journal of Molecular Sciences and look forward to hearing from you.
Yours sincerely and on behalf of all co-authors,
Martin Hoogduijn
m.hoogduijn@erasmusmc.nl
Answers to the questions and comments of reviewer #1
Reviewer 1:
- The authors need to include as a supplementary material the list of all the proteins detected in all the conditions tested.
Response: Upon acceptance of the manuscript, we will upload the raw data of the manuscript in a data repository. All data will be then publicly available.
Authors mentioned in the introduction that there bibliography describing proteome of MSC but especially in the case of MPs that are generated from MSCs It will be very valuable a comparative study between the whole cell and the MPs and after including EVs proteome study, therefore, they are vesicles naturally formed from MSCs.
Response: We agree with the reviewer that a comparison between the proteome of whole MSC and MP would have been very interesting. Such comparison could give an indication of the MSC proteins that are, and that are not incorporated into MP. The aim of our study was however to compare the proteome of MSC EVs with MSC MPs. To include proteomic analysis of whole MSC would mean a whole new study.
It will be valuable to include the full list of the 42 common proteins between MPs and EVs.
Response: This is a good point. We included a list containing the common and unique proteins in the four types of vesicles from the Venn diagram as Supplementary table 1.
In figure 4 at size 100% is impossible to read the code of the proteins.
Response: The file format of Figure 4 and all other figures containing plots have been changed to vector files (EPS), and this should ensure better visibility of the figures.
Authors claim that they demonstrate that the proteins in MP were functionally intact but they measured enzymatic activity of MP. They only measured ATPase activity and It will be valuable to include other activities related to other proteases or even the inhibitory activity of the MPs that it seems that authors detect some protease inhibitors in these vesicles.
Response: The ATPase activity measurements were included in the first place as a proof of principle to demonstrate that the proteins in MP are intact, and secondly to demonstrate that the proteomic detection of ATPases in MP, but not in EV, can be confirmed by enzyme activity measurements. We also detected ecto-nucleotidase activity in MP, but not in EV (Fig 6C). This correlates with the proteomic detection of ecto-nucleotidase in MP, but not in EV.
Authors talk about a more diverse proteome in MPs but they need to clarify that there are a lot of MPs, and that they contain a higher quantity of proteins than EVs.
Response: It is true that per MSC, we obtain more MP than EV. However, we loaded similar amounts of protein for mass spec. This allows us to state that within a fixed amount of protein, MP contain more types of protein than EV. We are sorry that the equal peptide loading was not mentioned in the manuscript. We included this information in methods section 4.7 Proteomic analysis. In the discussion (line 284-287) we indicated again that same amounts of peptide were loaded, and thus that MP contain more types of protein, while EV have high abundance of a smaller number of protein types.
Reviewer 2 Report
The manuscript by Tejeda-Mora et al reports an effort of the authors to characterize the protein content of the vesicles generated by osmotic lysis of mesenchymal stromal cells (MSCs), which they term “membrane particles (MPs)”, and compare it with that of extracellular vesicles (EVs). Tthe authors employed a straightforward, unbiased proteomics analysis for this study. However, I find it hard that this study shows any possibility of advancing our knowledge on both basic biology and therapeutic application.
- Most of all, I do not find any evidence in this manuscript that generating MPs can control the content, which the authors argue MPs seem superior to EVs, and which constitutes the premise of this study. I think the data just show that MPs enclose any proteins available to the proximity of cellular membrane. This may be why MPs contain such proteins as ribosomal proteins, cytoskeleton components, etc.
- A major finding of this manuscript is that the MP content is dissimilar to the EV content. Considering the procedural differences in preparing EVs and MPs, this is not surprising at all. What is the novel finding of this study?
- The authors showed that the yield of MPs is much higher than that of EVs. The description of mass spectrometry, however, does not indicate how the scale of the mass spec runs was normalized? Did they use a similar mass of protein prep, or a similar number of producing cells? It would be better describing the whole mass spec procedure in more detail in the main text.
- Also, it would be better showing the Venn diagram (Fig 3A) in scale, meaning that more proteins in a set should be presented with a bigger circle.
Author Response
Dear Dr. Albert Li,
We thank you for considering our manuscript for publication in the International Journal of Molecular Sciences and many thanks also to the reviewers who took time to read our work. The requested revisions are highlighted in the manuscript and we believe that these have strengthened the manuscript. Please find below a point-by-point description of the changes we made in the document and the answers to the questions raised by the reviewers.
We hope the revised manuscript is acceptable for publication in the International Journal of Molecular Sciences and look forward to hearing from you.
Yours sincerely and on behalf of all co-authors,
Martin Hoogduijn
m.hoogduijn@erasmusmc.nl
Answers to the questions and comments of reviewer #2
Reviewer #2:
Most of all, I do not find any evidence in this manuscript that generating MPs can control the content, which the authors argue MPs seem superior to EVs, and which constitutes the premise of this study. I think the data just show that MPs enclose any proteins available to the proximity of cellular membrane. This may be why MPs contain such proteins as ribosomal proteins, cytoskeleton components, etc.
Response: We completely agree with the reviewer that MP contain any protein that is associated with the membranes of the organelles they are composed of. What we intended to argue is that we can affect the protein expression of a cell with for instance IFNy treatment, and that when we subsequently generate MP of these cells, the impact of IFNy on the proteome of the cell can be found back in the MP. Figure 4C shows exactly this; IFNy treatment of MSC leads to a significant impact on the proteome of MP. Our statement that we can ‘control’ the protein content of MP is perhaps incorrect. A better term is ‘modify’. We use this wording in the abstract and in the final paragraph of the discussion.
In line 59, we changed the statement that ‘it is not possible to generate EV with controlled content and membrane protein composition’ to ‘it is unknown how to modify the content and membrane protein composition of EV’.
A major finding of this manuscript is that the MP content is dissimilar to the EV content. Considering the procedural differences in preparing EVs and MPs, this is not surprising at all. What is the novel finding of this study?.
Response: EV are the gold standard for MSC-derived vesicles. However, their therapeutic use is hampered by the low yield of EV and their contamination by soluble proteins. MP are a potential alternative. The novel finding is that we have now characterized MP in detail and that the proteome of MP is much richer than that of EV. MP appear to contain proteins from all cellular membranous compartments. This suggests that MP have partially different functions as EV.
The authors showed that the yield of MPs is much higher than that of EVs. The description of mass spectrometry, however, does not indicate how the scale of the mass spec runs was normalized? Did they use a similar mass of protein prep, or a similar number of producing cells? It would be better describing the whole mass spec procedure in more detail in the main text.
Response: The yield of MP is indeed much higher than that of EV. However similar amounts of protein were loaded in the mass spec runs, which allows a direct comparison between the mass spec data of MP vs EV. We added this information to paragraph 4.7
Also, it would be better showing the Venn diagram (Fig 3A) in scale, meaning that more proteins in a set should be presented with a bigger circle.
Response: Unfortunately it is mathematically not possible to have a Venn diagram of 4 sets with circles [1]. Similarly, when ellipses are used, scaling is not possible without recurring to non-symmetrical figures, which might lead to misleading visual representation of the data. However, to adjust the Venn diagram according to the reviewer’s comment, we updated the Venn diagram with a color scale to indicate which overlapping regions contain a lot, and which contain not so many proteins. This allows to visualize the numeric differences between subsets. The updated figure appears in Figure 3A.
References
1. J. Venn M.A. (1880) I. On the diagrammatic and mechanical representation of propositions and reasonings , Philosophical Magazine Series 5, 10:59, 1-18, DOI: 10.1080/14786448008626877
Round 2
Reviewer 1 Report
Congratulations
Reviewer 2 Report
The manuscript by Tejeda-Mora et al has addressed the issues that I raised before. Although I cannot still not completely accept the utility of MPs relative to EVs, the manuscript now appears much clearer and reads well. I applaud the authors for their efforts to conduct this study and to improve their manuscript.